# Sigma-1 Receptor Activation Improves Oligodendrogenesis and Promotes White-Matter Integrity after Stroke in Mice with Diabetic Mellitus

**DOI:** 10.3390/molecules28010390

**Published:** 2023-01-02

**Authors:** Wenjing Song, Yang Yao, Heling Zhang, Xin Hao, Liping Zhou, Zhixiao Song, Tiantian Wei, Tianyan Chi, Peng Liu, Xuefei Ji, Libo Zou

**Affiliations:** Department of Pharmacology, Shenyang Pharmaceutical University, 103 Wenhua Road, Shenhe District, Shenyang 110016, China

**Keywords:** stroke, diabetes, sigma-1 receptor, white-matter lesion, oligodendrocytes

## Abstract

Diabetes mellitus (DM) is a major risk factor for stroke and exacerbates white-matter damage in focal cerebral ischemia. Our previous study showed that the sigma-1 receptor agonist PRE084 ameliorates bilateral common-carotid-artery occlusion-induced brain damage in mice. However, whether this protective effect can extend to white matter remains unclear. In this study, C57BL/6 mice were treated with high-fat diets (HFDs) combined with streptozotocin (STZ) injection to mimic type 2 diabetes mellitus (T2DM). Focal cerebral ischemia in T2DM mice was established via injection of the vasoconstrictor peptide endothelin-1 (ET-1) into the hippocampus. Three different treatment plans were used in this study. In one plan, 1 mg/kg of PRE084 (intraperitoneally) was administered for 7 d before ET-1 injection; the mice were sacrificed 24 h after ET-1 injection. In another plan, PRE084 treatment was initiated 24 h after ET-1 injection and lasted for 7 d. In the third plan, PRE084 treatment was initiated 24 h after ET-1 injection and lasted for 21 d. The Y-maze, novel object recognition, and passive avoidance tests were used to assess neurobehavioral outcomes. We found no cognitive dysfunction or white-matter damage 24 h after ET-1 injection. However, 7 and 21 d after ET-1 injection, the mice showed significant cognitive impairment and white-matter damage. Only PRE084 treatment for 21 d could improve this white-matter injury; increase axon and myelin density; decrease demyelination; and increase the expressions of myelin regulator 2‘-3‘-cyclic nucleotide 3‘-phosphodiesterase (CNpase) and myelin oligodendrocyte protein (MOG) (which was expressed by mature oligodendrocytes), the number of nerve/glial-antigen 2 (NG2)-positive cells, and the expression of platelet-derived growth factor receptor-alpha (PDGFRα), all of which were expressed by oligodendrocyte progenitor cells in mice with diabetes and focal cerebral ischemia. These results indicate that maybe there was more severe white-matter damage in the focal cerebral ischemia of the diabetic mice than in the mice with normal blood glucose levels. Long-term sigma-1 receptor activation may promote oligodendrogenesis and white-matter functional recovery in patients with stroke and with diabetes.

## 1. Introduction

Ischemic stroke is the most common cause of death and long-term disability worldwide [1]. Diabetes mellitus (DM) is an independent risk for stroke recurrence [2,3]. Diabetes has been reported to increase incidence of ischemic stroke by two- to six-fold [3]. Diabetic patients have a higher risk of developing stroke and higher recurrence and mortality rates [4,5]. Diabetes exacerbates brain ischemia/reperfusion injury, which is mediated by enhanced endoplasmic reticulum (ER) stress and cell death [6,7].

White matter consists mainly of myelinated axons, oligodendrocytes (OLGs), and other glial cells and is highly vulnerable to ischemic injury [8]. Almost all instances of focal ischemia involve white matter [9]. Mature OLGs are responsible for forming and maintaining myelin sheaths, wrapping axons, providing nutrients to neurons, and promoting normal transmission of nerve centers [10]. OLGs are solely derived from differentiated OLG precursor cells (OPCs). Conversion of OPCs to OLGs is a highly sensitive and complex process [11]. White-matter injury following stroke is closely related to OLG demyelination, axonal injury, and eventually neurological deficits. It also induces proliferation of pre-existing OPCs, which restore OLGs and subsequently resheath. However, most OPCs fail to differentiate into mature OLGs, resulting in insufficient remyelination and unsuccessful white-matter repair [10,11,12]. Therefore, promoting oligodendrogenesis may represent a potential therapeutic strategy to increase white-matter repair and neurological recovery after stroke.

The sigma-1 receptor is an ER chaperone protein that is upregulated during ER stress and regulates calcium homeostasis [13]. It is widely expressed in the central nervous system (CNS) and is involved in neurological disease [14,15]. The sigma-1 receptor colocalizes with specific markers of progenitor and mature OLGs [16]. Sigma-1 receptor agonists protect OLGs and OPCs from apoptosis, excitotoxicity, and inflammation in multiple sclerosis [17]. Treating rat OLG progenitors (CG-4 cell line) with sigma-1 receptor agonists or overexpression of sigma-1 receptors enhances CG-4 cell differentiation, whereas Sig-1R siRNA inhibits differentiation [18]. 

Our previous study showed that in sigma-1 receptor knockout mice, brain ischemia–reperfusion produces serious immunoglobulin (Ig) G leakage and degradation of blood–brain barrier (BBB) structural proteins [19]. The sigma-1 receptor agonist PRE084 can ameliorate bilateral common-carotid-artery occlusion-induced cognitive impairment, neuronal apoptosis, and ER stress in mice [20,21]. However, the effects of PRE084 and sigma-1 receptors on white-matter damage in focal cerebral ischemia remain unclear. Therefore, we used type 2 diabetes mellitus (T2DM) mice to aggravate endothelin-1 (ET-1)-induced focal cerebral ischemia damage in the brain. Finally, we evaluated the effects of PRE084 on oligodendrogenesis and white-matter function.

## 2. Results

### 2.1. Sigma-1 Receptor Activation Reverses Cognitive Dysfunction and Improves Long-Term Histological Deficits in Mice with Diabetes and Stroke

The blood glucose and weight of the diabetic mice were significantly higher than those of sham mice without diabetes, and the corresponding fasting blood sugar levels were higher than 11.1 mmol/L (Figure 1B,C). The Y-maze test is a spatial working memory test. There were no significant differences in alternation behavior between the groups 1 d after ET-1 injection. Both the nondiabetic and diabetic mice with stroke showed significant spatial working memory impairment 7 d after ET-1 injection. Alternation behavior was significantly decreased in both the nondiabetic and diabetic mice with stroke 21 d after ET-1 injection. Stroke performance in the diabetic mice was attenuated with PRE084 (a sigma-1 receptor agonist) treatment (Figure 1D). A novel object recognition test was used to evaluate recall and visual-recognition impairments. At 21 d after ET-1 injection, the preferential index (PI) and discrimination index (DI) were significantly decreased in the diabetic mice with stroke; this was attenuated with PRE084 treatment (Figure 1E,F). The passive avoidance test was used to evaluate long-term memory and learning function. We found that PRE084 treatment had no effect on error time or escape latency in stroke in the diabetic mice 21 d after ET-1 injection (Figure 1F,G). All of the behavioral results indicated that the mice with diabetes and stroke suffered more severe cognitive deficits than did the nondiabetic mice with stroke. Long-term PRE084 treatment has therapeutic potential for improving the neurological function of diabetic mice with stroke.

Neuronal injuries were measured using Nissl staining. Varying degrees of atrophy of the Nissl bodies appeared 1, 7, and 21 d after ET-1 injection (Figure 2). Stroke in the diabetic mice caused more severe neuronal damage than did stroke in the nondiabetic mice, especially at 7 and 21 d after ET-1 injection in the hippocampal CA1 region and the cerebral cortex. PRE084 improved neuronal injury 21 d after ET-1 injection (Appendix A). These results show that diabetes aggravates nerve injury caused by stroke, and activating the sigma-1 receptor can ameliorate this nerve damage.

### 2.2. Sigma-1 Receptor Activation Prevents Levels of Brain Demyelination and White-Matter Damage in Mice with Diabetes and Stroke

White-matter injury and myelin loss are key components of the progression of neuronal injury after stroke. Therefore, the myelination score was evaluated using the expression of myelin basic protein (MBP) and Luxol-fast-blue staining (LFB) in the perilesion corpus striatum (CS) and the peri-infarct corpus callosum (CC). At 7 and 21 d after ET-1 injection, both the nondiabetic and diabetic mice with stroke showed a loss of myelin in the striatum and corpus callosum (Figure 3). PRE084 prevented myelin loss 21 d after ET-1 injection (Figure 3). PRE084 treatment also significantly increased MBP expression (Figure 4A,B,F).

### 2.3. Sigma-1 Receptor Activation Promotes Mature Oligodendrogenesis and Myelin Regeneration in Mice with Diabetes and Stroke

Mature oligodendrogenesis is responsible for myelin formation and maintenance. At 7 and 21 d after ET-1 injection, both the nondiabetic and diabetic mice with stroke showed a significant decrease in expressions of mature OLG markers, including myelin regulator 2‘,3‘-cyclic nucleotide 3‘-phosphohydrolase (CNpase) (Figure 4C,F and Figure 5) in the hippocampus and the cerebral cortex, myelin-associated glycoprotein (MAG) (Figure 4E,H), and myelin OLG protein (MOG) (Figure 4D,G) in the hippocampus. However, no significant decrease in protein expression was observed 1 d after ET-1 injection (Appendix A). We found that PRE084 only enhanced the expressions of CNpase, MAG, and MOG at 21 d after ET-1 injection, not at 7 d after. These results showed that the activation of the sigma-1 receptor promoted myelin regeneration via increasing the mature OLGs in the brains of the diabetic mice with stroke.

### 2.4. Sigma-1 Receptor Activation Promotes Oligodendrogenesis Differentiation in Mice with Diabetes and Stroke

Olig2 is a transcription factor expressed in all OLG-lineage cells regardless of maturation state. the Myelin gene regulatory factor (Myrf) is an essential transcription factor for oligodendrocyte differentiation. We quantified temporal changes in the positive areas of Olig2 and Myrf in the CA1 and the SL–SE hippocampal layers from 1 to 21 d after ET-1 injection. As expected, we found that the number of Olig2- and Myrf-positive cells increased in both the nondiabetic and diabetic mice with stroke at 7 and 21 d in peri-ischemic infarction. PRE084 treatment decreased the amount of Olig2- and Myrf-positive cells at 21 d, but not at 7 d, after ET-1 injection, which indicates that the activation of the sigma-1 receptor promoted the conversion of the OLGs to remyelination (Figure 6 and Figure 7).

### 2.5. Sigma-1 Receptor Activation Regulates OPC Proliferation in Mice with Diabetes and Stroke

Neuron–glial antigen 2 (NG2) cells are glial cells that tilt throughout the gray and white matter of the CNS. Platelet–derived growth factor receptor-α (PDGFα) is a tyrosine kinase receptor on the cell surface and an oligodendrocyte progenitor cell marker. NG2 glia and PDGFRα are activated following brain injury. Previous studies have suggested active and functional roles for NG2 glia and PDGFRα in the CNS. In accordance with previous studies, we found that the number of OPCs was significantly increased in both the nondiabetic and diabetic mice, with stroke at 7 and 21 d after ET-1 injection in the hippocampal CA1 region and the SL–SE hippocampal layers. Stroke in the diabetic mice showed more NG2-positive cells than did stroke in the nondiabetic mice 21 d after ET-1 injection in the hippocampal CA1 region, but not PDGFRα. PRE084 treatment significantly decreased the amounts of NG2 and PDGFRα at 21 d after ET-1 injection. This result indicates that maybe activating the sigma-1 receptor can regulate OPC proliferation and even promote conversion of OPCs to OLGs, but this conclusion needs much more research to be confirmed (Figure 8 and Figure 9).

## 3. Discussion

In the present study, we demonstrated for the first time that the sigma-1 receptor agonist PRE084 exerts neuroprotective effects via white-matter repair. The findings of this study are as follows: First, neurological dysfunction, white-matter damage, and myelin loss occurred on days 7 and 21 of ET-1 injection-induced focal cerebral ischemia, though no significant injury was observed on day 1. Second, the level of LFB staining and the MBP expression were lower in the diabetic mice with stroke than in the nondiabetic mice with stroke, but with no significance. Third, long-term PRE084 treatment (for at least 21 d) could prevent diabetes-aggravated white-matter damage via increased myelin regeneration and promotion of neuroprotection. However, in our previous study, PRE084 showed an antiapoptosis effect in mice with cerebral ischemia/reperfusion injury [21]. Therefore, sigma-1 receptor activation may play a role in protecting OLGs from apoptosis other than increasing myelin regeneration. A lack of transmission electron microscopy of the myelin sheath at different conditions and time points is the main defect of this study.

ET-1 is a 21-amino-acid peptide with strong vasoconstrictive properties. Vasoconstriction-induced via ET-1 is one of the most commonly used models of ischemic stroke [22]. The main advantages of this model include enablement of performing the procedure quickly and of controlling artery constriction via alteration of the dose of ET-1 delivered [23,24,25]. Some studies have found that size of the ischemic focus is positively correlated with dose of ET-1, accompanied by a small amount of reperfusion. Severe cognitive disability has also been noted in the open-field test, the novel object recognition test [26], and the Morris water maze test [27]. Our observations are in accordance with previous studies that have shown that hippocampal injection of ET-1 caused spatial working memory dysfunction, recall memory dysfunction, and long-term memory dysfunction in the Y-maze, novel object recognition, and passive avoidance tests. Thus, ET-1 injection mimicked cognitive impairment caused by focal cerebral ischemia in this study. Previous studies have reported that diabetes can aggravate cerebral ischemic injury both in the early reperfusion stage [28] and in the late reperfusion stage of cerebral ischemia [29]. In this study, diabetes aggravated neuronal damage after 7 and 21 d of focal cerebral ischemia (bilateral hippocampal ischemia), but not after 1 d. This may be a characteristic of the ET-1-induced ischemic model, in which it is difficult to control spontaneous reperfusion. The duration of vasoconstriction reperfusion for ET-1 induction may have been different from that of human stroke, with reperfusion over a period of hours to days following occlusion [30,31]. The sigma-1 receptor agonist PRE084 has been shown to protect against neuronal damage in stroke via multiple mechanisms; sigma-1 receptor activation regulates brain-derived neurotrophic factors through the NR2A-CaMKIV-TORC1 pathway [20,32], prevents stress-mediated ER apoptosis [21], and protects the BBB [19] in the bilateral common-carotid-artery occlusion-induced global ischemia-reperfusion model. Our study found that PRE084 also has a neuroprotective effect on cerebral ischemia via reparation of white-matter damage and enhancement of remyelination. 

White-matter injury is a key component of poststroke neuronal injury. Diabetes can aggravate white-matter injury after cerebral ischemia [33,34,35]. White matter is composed mainly of myelinated axons; thus, white-matter damage is manifested as myelin loss and axon injury, resulting in impairment of impulse transmission of neurons, which is also related to long-term cognitive dysfunction after cerebral ischemia [36]. In this study, at 7 and 21 d after stroke, both the nondiabetic and diabetic mice with stroke showed obvious myelin loss (LFB-positive staining and expression of MBP decreased) in peri-infarct areas, such as the corpus callosum and the striatum. The diabetic mice with stroke suffered more severe damage than did the nondiabetic mice with stroke. White matter is fragile [8], and it is important to repair it with myelin regeneration and oligodendrogenesis enhancement [11,37]. Unfortunately, cerebral stroke has limited the capacity for remyelination, at least in part due to the failure of OPC differentiation into mature, myelinating OLGs, as well as focal cerebral stroke with diabetes. Thus, promotion of OPC differentiation may facilitate axonal remyelination, white-matter repair, and long-term neurological recovery in patients with stroke [38]. Our results showed that PRE084 could attenuate white-matter injury at 21 d after focal cerebral ischemia, even if it were aggravated by diabetes. This result also explains why diabetes worsens the clinical prognosis of cerebral stroke and makes it more difficult to cure.

Oligodendrogenesis is the process through which OPCs generate mature OLGs [39]. OPCs are found throughout the CNS, in the white and gray matter, and are the fourth glial cell type. They are sensitive to pathological conditions and migrate to infarct areas rapidly to promote OLG production and myelin repair [40,41,42]. Myelin loss is due to the inability of mature OLGs to differentiate after ischemic injury and thus to produce nonfunctional myelin. In this study, we found that 7 and 21 d after stroke, both the nondiabetic and diabetic mice showed a continuous increase in their numbers of NG2- and PDGFα-positive cells and a decrease in mature OLGs. The diabetic mice with stroke had more NG2- and PDGFα-positive cells than did the nondiabetic mice with stroke 21 d after ET-1 injection. This also explains why diabetes aggravates cerebral ischemic injury. The presence of OLG precursors in the mature brain may provide an opportunity for replenishment of OLGs after injury [43,44]. PRE084 reduced NG2 levels after 21 d of treatment in the diabetic mice with stroke, which suggests that activation of the sigma-1 receptor regulated OPC proliferation. According to our results, it was difficult to judge whether ET-1 induced more severe demyelination or slowed the remyelination process. Ischemic stroke leads to white-matter injury; inhibition of demyelination or promotion of remyelination may have therapeutic value in clinical practice. Remyelination is an important repair mechanism triggered after an ischemic-stroke-induced white-matter injury, but it often fails due to a lack of recruitment of OPCs to the demyelinated area and inadequate differentiation of OPCs [45]. In our study, we found that sigma-1 receptor activation could promote OLG differentiation and OPC proliferation in mice with diabetes and stroke. We estimate that demyelination and remyelination may be both involved in the toxicity of ET-1 or the neuroprotective quality of PRE084. 

Above all, long-term sigma-1 receptor activation may promote oligodendrogenesis and white-matter functional recovery in diabetic mice with stroke.

## 4. Materials and Methods

### 4.1. Animals

Male C57BL/6 mice (8-week-old) were purchased from Liao Ning Chang Sheng Biotechnology Co. Ltd. (Shenyang, China). These animals were housed in a specific pathogen-free facility (temperature, 23 ± 1 °C; relative humidity, 50 ± 5%) with a 12 h light/dark cycle. All of the animal studies were performed in strict accordance with the Chinese legislation on use and care of laboratory animals and the guidelines established by the Institute for Experimental Animals at Shenyang Pharmaceutical University (SYPU-IACUC-GZR2021-03.01-096).

### 4.2. Induction of Hyperglycemia

The mice were randomly divided into two groups: a control group (normal diet: 67% carbohydrates, 12% fat, and 21% protein) and a high-fat diet (HFD) group (20% carbohydrates, 60% fat, and 20% protein). After 5 wks. of HFD feeding, the mice in the HFD group were intraperitoneally injected with STZ (40 mg/kg) (Sigma-Aldrich, USA) once a day for 5 d. Mice in the control group were injected with a citric acid buffer [46]. To validate the successful establishment of the T2DM model, the fasting blood glucose levels in the mice were assessed. Only mice with fasting blood glucose levels higher than 11.1 mmol/L were used in the follow-up study.

### 4.3. Bilateral Hippocampal Injection of ET-1

The mice were anesthetized with 2.5% avertin (Sigma-Aldrich, St Louis, MO, USA). The fur was then removed from each scalp, and each head was held in a stereotaxic apparatus. The vasoconstrictor ET-1 (1 μg/μL, Genscript, Nanjing, China) was injected into the hippocampus at the following coordinates: AP, −2.06 mm; ML, ±1.30 mm; and DV, −2.00 mm. The infusion rate was 0.5 μL/min. The needle was left in place for 5 min before retraction and suturing. Mice in the sham and diabetic groups were injected with the same volume of saline [47,48]. All of the mice were returned to their home cages for a duration determined through experimental conditions.

### 4.4. Drug and Treatment Schedule

Mice with fasting blood glucose > 11.1 mmol/L were used in this study and subjected to a focal cerebral ischemia stroke model via bilateral hippocampal injection of ET-1. T2DM mice were divided into five groups of 9–10 mice each: (1) the non-ET-1 injection and nondiabetes group (sham); (2) the non-ET-1 injection and diabetes group (D2TM); (3) the ET-1 injection and nondiabetes group (ET-1); (4) the ET-1 injection and diabetes group (D2TM+ET-1); and (5) the ET-1 injection, diabetes, and PRE084 treatment group (D2TM+ET-1+PRE084). PRE084 (Medchem Express, Monmouth Junction, New Jersey, USA) at a dose of 1 mg/kg was injected intraperitoneally (i.p.) into each mouse once daily until the mice were sacrificed. The PRE084 dose selection was based on our previous study [49]. Three different treatment plans were used in this study. 

Experiment A: Treatment with 1 mg/kg PRE084 (i.p.) for 7 d before ET-1 injection. The Y-maze test was used to evaluate working memory dysfunction 24 h after ET-1 injection. The mice were then sacrificed, and their brain tissue was dissected.

Experiment B: The PRE084 treatment began 24 h after ET-1 injection and lasted for 7 d. The Y-maze, novel object recognition, and passive avoidance tests were used to assess cognitive dysfunction. After the behavioral test, the mice were sacrificed, and their brain tissue was dissected.

Experiment C: The PRE084 treatment began 24 h after ET-1 injection and lasted 21 d. The Y-maze, novel object recognition, and passive avoidance tests were used to assess cognitive dysfunction. After the behavioral test, the mice were sacrificed, and their brain tissue was dissected. The experimental schedule is shown in Figure 1.

### 4.5. Y-Maze Test 

According to our previous report [50], the instrument consisted of three arms (length, 40 cm; width, 10 cm; and height, 12 cm). The experiment began with the mice facing away from one arm; they were allowed to explore the device freely for 5 min. The total number of arm entries (n) and the sequence of entries were recorded. The mice entered three different arms in succession, and a correct alternation response was recorded. The ratio of spontaneous alternation reaction could be obtained through computation of the number of successful alternations/(total number of entries − 2) × 100%.

### 4.6. Novel Object Recognition Test

The novel object recognition test was performed as described in our previous study [50]. The instrument consisted of a square box (length 50 cm × width 50 cm × height 15 cm). The experiment was divided into adaptation and test stages. On the first 2 d of the adaptation stage, the mice were placed with their backs to the site and allowed to explore freely for 3 min in the open field, twice per day. On the third day of the test stage, only one mouse was added at a time, and two identical objects (A1 and A2) were placed equidistantly from the open field after 3 min of free exploration. The mice were then placed at the same distance from the two objects, with their backs to the field. The time they spent exploring the two objects within 5 min was recorded, and the mice were returned to the cage after exploration. After 1 h, another new object, B, replaced A2 at the same location. The exploration time for each object was recorded during the training and retention sessions. The PI and DI values were calculated (PI = [time spent exploring the novel object/total exploration time], DI = [(time spent exploring the novel object time spent exploring the familiar object)/total exploration time]).

### 4.7. Passive Avoidance Test

The passive avoidance test was performed as previously described [50]. The experimental apparatus was composed of bright and dark rooms. The bright room had strong light irradiation. In the dark room, the copper grid passed through an alternating current (36 V). The experiment was divided into 2 d: The first day was the training phase, wherein the mice were individually placed in a hole without electricity in the bright room and left to roam freely for 3 min. An alternating current was then applied as the mice were placed in the dark room. Thus, the normal reaction of the mice was to run back to the bright room to avoid electric shock. The number of times the mice entered the dark room again within 5 min after a shock was recorded as “error times.” The method used during the test phase was the same used during the training-phase day; the former was performed after a 24 h retention interval.

### 4.8. LFB Staining and Nissl Staining

LFB staining and Nissl staining were performed as described previously [51,52]. In order to detect morphological changes of neurons and demyelination, brain slices were placed in 0.1% LFB (Solarbio, Beijing, China) at room temperature overnight before being placed in 95% alcohol. Afterward, they were differentiated with 0.05% lithium carbonate, color-separated using 70% alcohol, dehydrated using graded ethanol, vitrified using dimethylbenzene, and deposited in neutral gum mountant.

These slices were placed in Nissl (Meilunbio, Dalian, China) for 1 h at 60 °C, followed by 95% alcohol. They were then washed with distilled water, then placed again in 70% alcohol. Finally, xylene was added and the tablet was sealed.

### 4.9. Immunohistochemistry

Immunohistochemistry was performed as previously described [53]. Half of the brain tissue of all of the mice was obtained via perfusion with paraformaldehyde, and the brain tissue was fixed, dehydrated, and embedded in paraffin. The sections were incubated with mouse anti-CNpase (ab6319, Abcam, Cambridge, UK), rabbit anti-PDGFRα (bs-0231R, Bioss, Beijing, China) and the MYRF (A16355, ABclonal, Wuhan, China), at 4 °C overnight. After the antibodies were washed with PBS, sections were incubated with biotin-labeled goat antimouse antibodies (1:200; Boster, Wuhan, China) at 37 °C for 60 min. DAB was used to mark positive signals. The intensity of each section was quantified using ImageJ software.

### 4.10. Immunofluorescence

Immunofluorescence was performed as previously described [54]. To test the levels of OPCs and OLGs in diabetes with focal cerebral ischemia, OPCs that used rabbit anti-NG2 (DF12589, Affinity, China) and OLGs with rabbit anti-Olig2 (65915, Cell Signaling Technology, Danvers, MA, USA) were stored at 4 °C overnight. After we washed the antibodies with PBST, we incubated slices with biotin-labeled goat antirabbit antibodies (1:200, SA00013-4, Solarbio, Beijing, China) at 37 °C for 60 min. The slices were again cleaned with PBST, and DAPI was added. The fluorescence quencher was stored to prevent quenching.

### 4.11. Western Blotting

The hippocampus and cerebral cortex tissues were homogenized and centrifuged, and supernatants were extracted. All of the samples were quantitatively analyzed using BCA for total protein content. The 25 ug of protein was run on a 10% gradient SDS–PAGE gel and electrophoretically transferred to polyvinyl difluoride (PVDF) membranes (Millipore, Billerica, MA, USA). Nonspecific binding proteins were blocked in 5% milk for 1.5 h at room temperature. Incubated protein bands with primary antibodies were left overnight at 4 °C before the membranes were incubated with antimouse IgG or antirabbit IgG antibodies for 2 h at room temperature; the antibodies were horseradish-peroxidase-labeled. Blots were visualized using an ECL western blotting kit (Advansta). Protein bands were scanned and semiquantified with densitometry using ImageJ software. The primary antibodies used were MBP (10458-1-AP, Proteintech, Wuhan, China), CNpase (ab6319, Abcam, Cambridge, UK), MAG (DF7669, Affinity, Liyang, China), and MOG (12690-1-AP, Proteintech, Wuhan, China).

### 4.12. Statistical Analysis

Data are presented as mean ± SD or SEM (for blood sugar levels and behavioral experiments). Differences between any two groups were analyzed using one-way analysis of variance (ANOVA) followed by Fisher’s least significant difference (LSD) multiple comparisons test with homogeneity of variance, or Dunnett’s T3 test with heterogeneity of variance. A value of *p* < 0.05 was considered to indicate significance. All analyses were performed using SPSS 17.0.

## Figures and Tables

**Figure 1 molecules-28-00390-f001:**
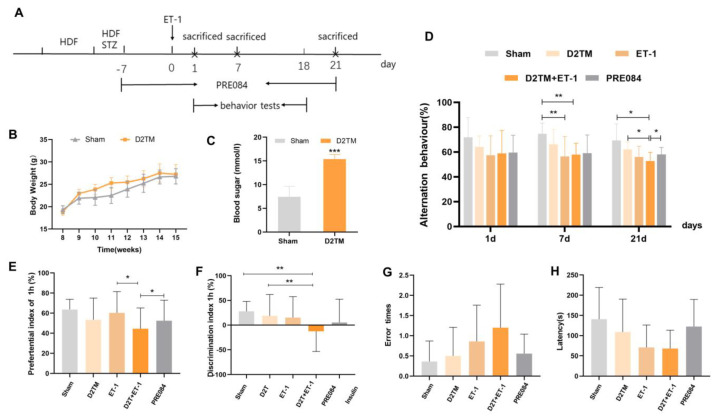
PRE084 improved long-term histological deficits and reversed cognitive dysfunction after stroke in diabetic mice. (**A**) The experimental design, PRE084, is a Sig1R agonist. (**B**) The body weights of the D2TM mice were higher than those of the sham mice. (**C**) The blood sugar levels of the D2TM mice were significantly higher than those of the sham mice. *n* = 60–70 in the sham group and n = 90–110 in the D2TM group, with *** *p* < 0.001 vs. sham. (**D**) Spatial working memory tests were evaluated using the Y-maze test. Alternation behavior was recorded after 1, 7, and 21 d of stroke. (**E**,**F**) Recall memory was evaluated via the novel object recognition test. Preferential and discrimination indices were recorded after 21 d of stroke. (**G**,**H**) Long-term memory was evaluated via the passive avoidance test. The error time and escape latency were recorded after 21 d of stroke. Data are expressed as mean ± SEM; *n* = 8–10/ group. * *p* < 0.05, ** *p* < 0.01, *** *p* < 0.001.

**Figure 2 molecules-28-00390-f002:**
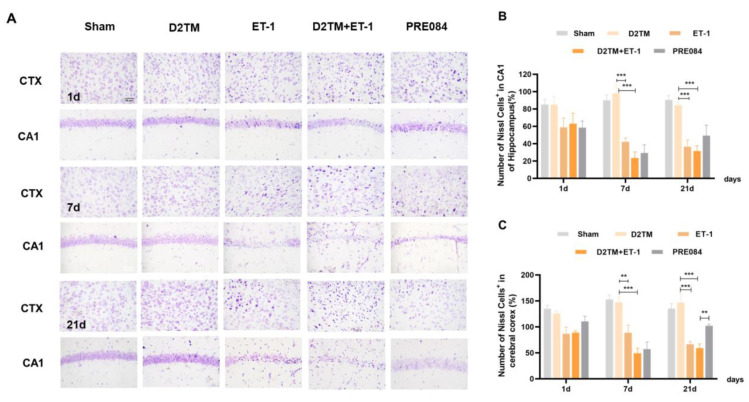
PRE084 attenuated neuronal damage in diabetic mice with stroke. (**A**) Nissl staining of the surviving cells in the hippocampal CA1 region and the cerebral cortex at 1, 7, and 21 d after ET-1 injection. (**B**,**C**) Quantification of the nissl positive cell number in the hippocampal CA1 region and the cerebral cortex. *n* = 4–5/group. Scale bar = 50 µm. ** *p* < 0.01, *** *p* < 0.001.

**Figure 3 molecules-28-00390-f003:**
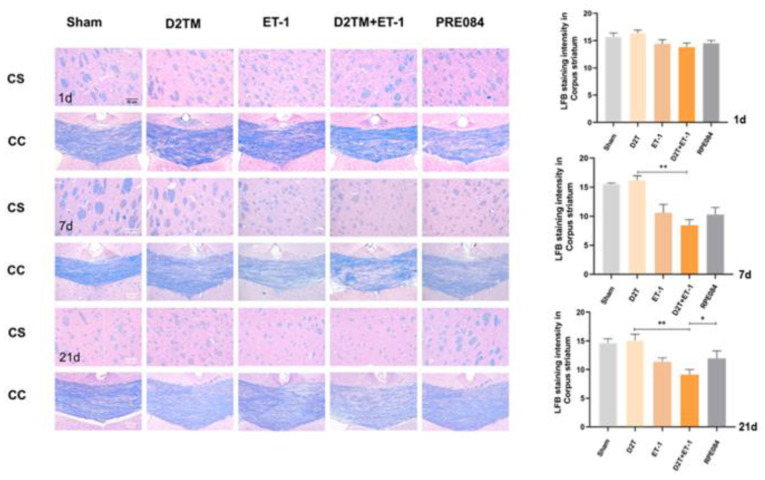
PRE084 prevented levels of brain demyelination and white-matter damage in mice with diabetes and stroke. Representative images of Luxol-fast-blue staining (LFB) in the peri-infarct corpus striatum (CS) and the corpus callosum (CC) 1, 7, and 21 d after ET-1 injection. Quantitative analysis of myelin density was expressed as the percentage of the immunostaining intensity in the CS. Data are expressed as mean ± SEM; *n* = 4–5/group. Scale bars = 50 μm. * *p* < 0.05, ** *p* < 0.01.

**Figure 4 molecules-28-00390-f004:**
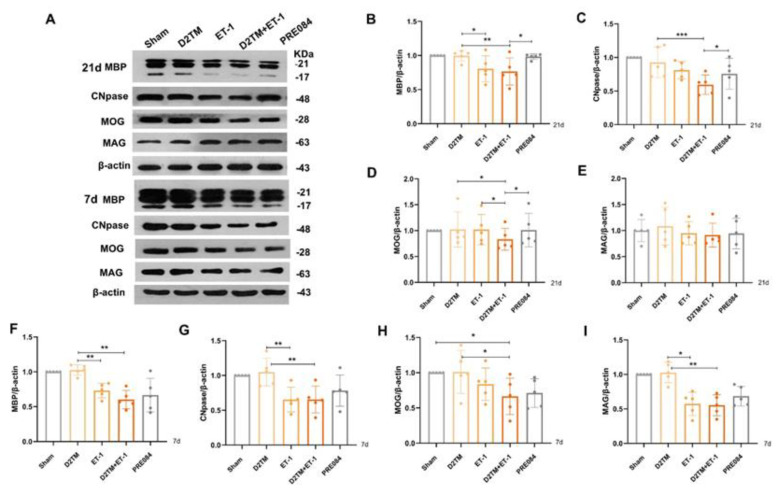
PRE084 promoted mature oligodendrogenesis and myelin regeneration in diabetic mice with stroke. (**A**) An immunoblot analysis of the hippocampus with antibodies against MBP, CNpase, MOG, and MAG at 21 and 7 d after ET-1 injection. (**B**–**E**) Quantification of the band densities of MBP, CNpase, MOG, and MAG at 21 d after ET-1 injection. (**F**–**I**) Quantification of the band densities of MBP, CNpase, MOG, and MAG at 7 d after ET-1 injection. Data are expressed as mean ± SEM; *n* = 5/group. * *p* < 0.05, ** *p* < 0.01, *** *p* < 0.001.

**Figure 5 molecules-28-00390-f005:**
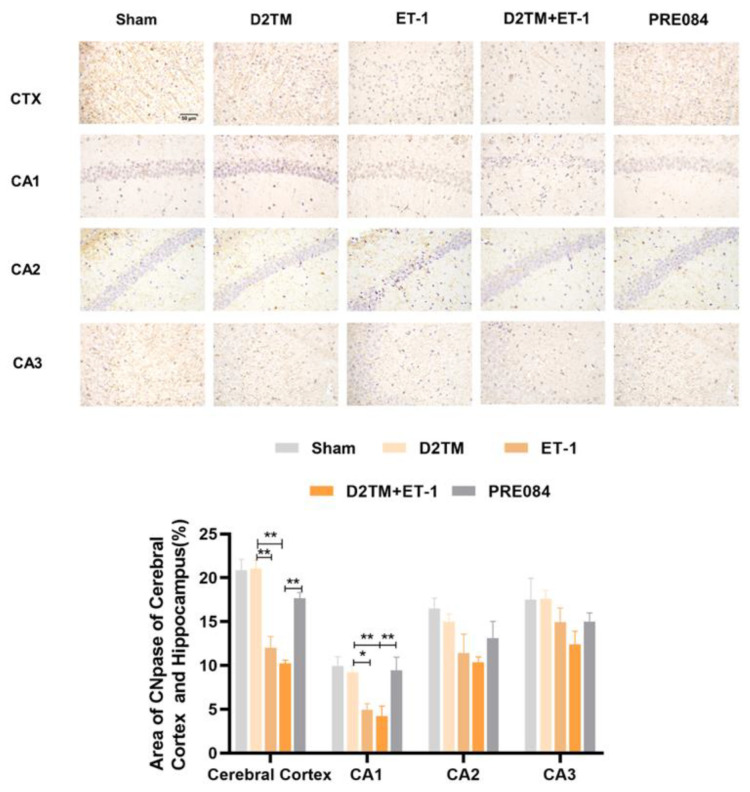
PRE084 increased the expression of CNpase in the cerebral cortex and in the hippocampal CA1, CA2, and CA3 regions of diabetic mice with stroke. Top: Representative images of CNpase 21 d after ET-1 injection. Bottom: Quantification of the area of the CNpase-immunostained cells in the cerebral cortex and in the hippocampal CA1, CA2, and CA3 regions. Data are expressed as mean ± SEM; *n* = 4–5/group. Scale bars = 50 μm. * *p* < 0.05, ** *p* < 0.01.

**Figure 6 molecules-28-00390-f006:**
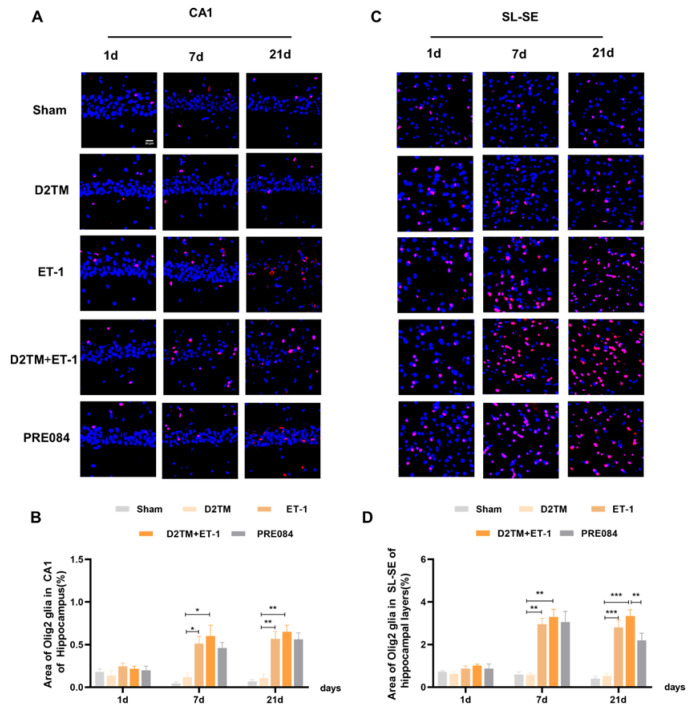
Sigma-1 receptor activation promoted oligodendrogenesis differentiation in diabetic mice with stroke. (**A**,**C**) Olig2 immunostaining (red) with DAPI (blue) was performed to detect oligodendrogenesis in the CA1 and the substratum lacunosum (SL) and substratum eumoleculare (SE) hippocampal layers at 1, 7, and 21 d after ET-1 injection. (**B**,**D**) Quantification of the area of the Olig2-immunostained cells in the CA1 and SL–SE of the hippocampus. Data are expressed as mean ± SEM; *n* = 4–5/group. Scale bars = 25 μm. * *p* < 0.05, ** *p* < 0.01, *** *p* < 0.001.

**Figure 7 molecules-28-00390-f007:**
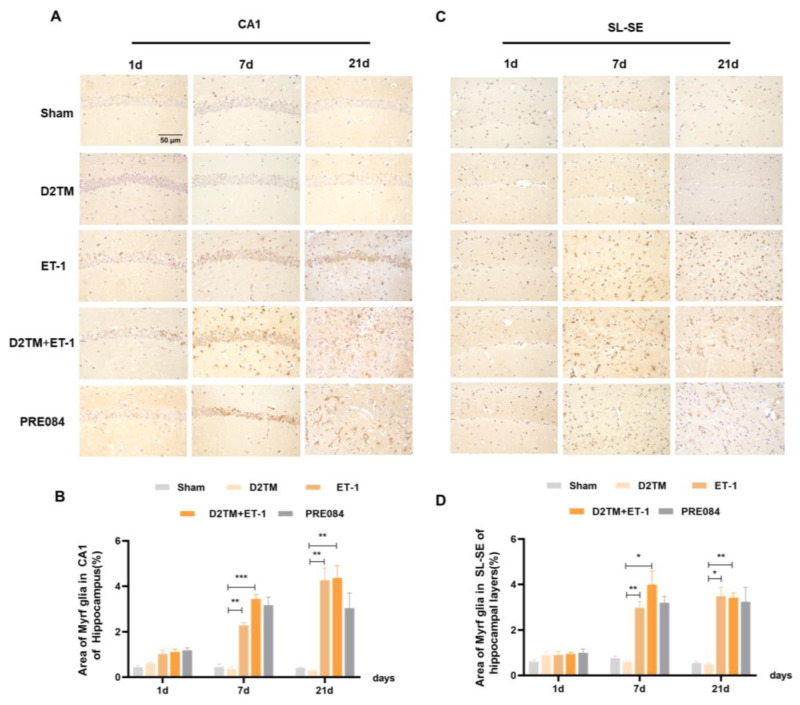
Sigma-1 receptor activation regulated Myrf expression in stroke in diabetic mice. (**A**,**C**) Myrf immunohistochemistry was performed to detect oligodendrogenesis in the CA1 and the SL–SE hippocampal layers at 1, 7, and 21 d after ET-1 injection. (**B**,**D**) Quantification of the area of the Myrf-immunostained cells in the CA1 and SL–SE of the hippocampus. *n* = 4–5/group. Scale bars = 50 μm. * *p* < 0.05, ** *p* < 0.01, *** *p* < 0.001.

**Figure 8 molecules-28-00390-f008:**
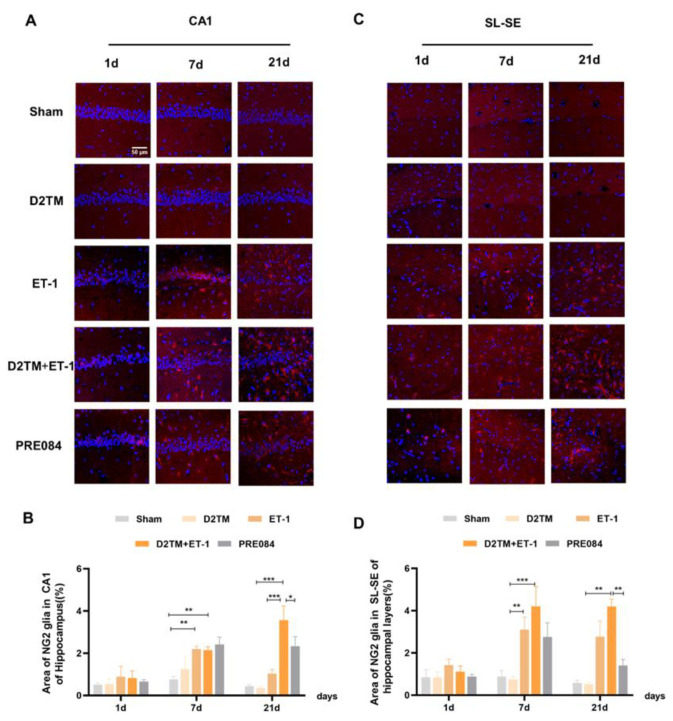
PRE084 regulated OPC proliferation in mice with diabetes and stroke. (**A**,**C**) NG2 immunostaining (red) with DAPI (blue) was performed to detect oligodendrocyte progenitor cells in the CA1 and the SL–SE hippocampal layers at 1, 7, and 21 d after ET-1 injection. (**B**,**D**) Quantification of the area of the NG2-immunostained cells in the CA1 and SL–SE of the hippocampus. Data are expressed as mean ± SEM; *n* = 4–5/group. Scale bars = 25 μm. * *p* < 0.05, ** *p* < 0.01, *** *p* < 0.001.

**Figure 9 molecules-28-00390-f009:**
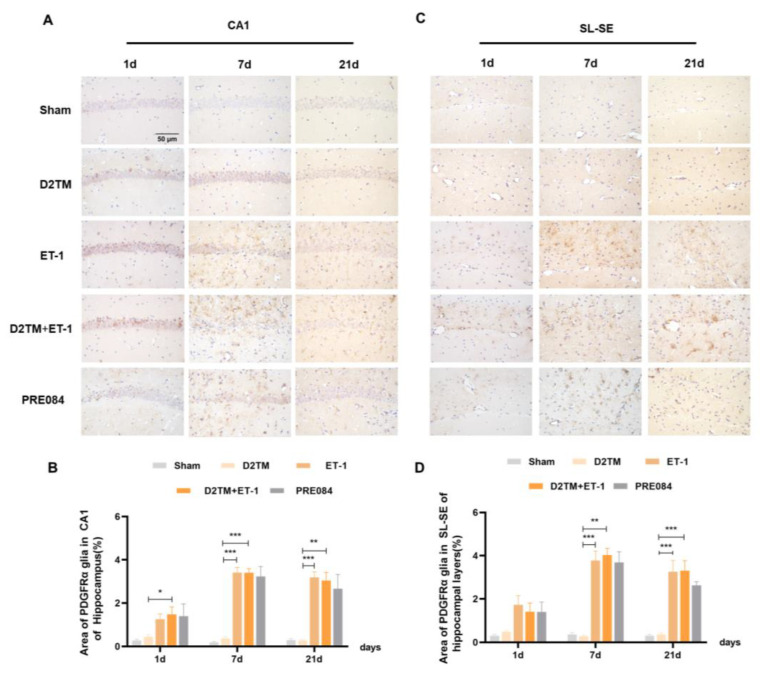
PRE084 regulated the expression of PDGFRα in stroke in diabetic mice. (**A**,**C**) PDGFRα immunohistochemistry was performed to detect oligodendrocyte progenitor in the CA1 and the SL–SE hippocampal layers at 1, 7, and 21 d after ET-1 injection. (**B**,**D**) Quantification of the area of the PDGFα-immunostained cells in the CA1 and SL–SE of the hippocampus. *n* = 4–5/group. Scale bars = 50 μm. * *p* < 0.05, ** *p* < 0.01, *** *p* < 0.001.

## Data Availability

Original data supporting this research are available from the authors.

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
