# Peer review of "Sigma-1 Receptor Activation Improves Oligodendrogenesis and Promotes White-Matter Integrity after Stroke in Mice with Diabetic Mellitus"

_molecules, 2023, doi:10.3390/molecules28010390_

Round 1

Reviewer 1 Report

This manuscript is well written and provides a meaningful perspective by targeting myelin regeneration after stroke. In this paper, authors draw three major conclusions.

First, neurological dysfunction, white matter damage, and myelin loss occurred on days 7 and 21 of ET-1 injection-induced focal cerebral ischemia, though no significant injury was observed on day 1. This part is well established and has solid evidence.

“Second, diabetic mice suffered more severe white matter damage and neurological dysfunction after ET-1 injection than did non-diabetic mice.” Diabetic mice suffer more sever white matter damage was concluded without strong and direct evidence. In figures 3 and 4, LFB staining and myelin protein level show no significant difference at 7days. The difference found in 21 days may come from slower remyelination. In diabetes mice, RT-1 injection induced more sever demyelination or slower the remyelination process is worth more clear analysis.

“Third, long-term PRE084 treatment (for at least 21 d) could prevent diabetes aggravated white matter damage by increasing myelin regeneration and promoting neuroprotection. “  Different oligodendrocyte linage cells number or protein level change are good sign of myelin recovery. Electron microscopy of myelin should be added to give a clear conclusion about myelin recovery. The last thing is that the authors gave PRE084 before ET-1 injection, so the drug may also play a role in protecting OL from apoptosis not only by increasing myelin regeneration. EM of myelin sheath at different conditions and time point will help the authors answer these questions.

Author Response

Paper: molecules-2039931

Title: Sigma-1 receptor activation improves oligodendrogenesis and promotes white matter integrity after stroke in mice with diabetic mellitus

Thank you very much for your comments on our paper. We have revised our paper according to your comments. All the revisions were emphasized in red.

Reviewer 1:

1.“Second, diabetic mice suffered more severe white matter damage and neurological dysfunction after ET-1 injection than did non-diabetic mice.” Diabetic mice suffer more sever white matter damage was concluded without strong and direct evidence. In figures 3 and 4, LFB staining and myelin protein level show no significant difference at 7days. The difference found in 21 days may come from slower remyelination. In diabetes mice, ET-1 injection induced more sever demyelination or slower the remyelination process is worth more clear analysis.

The authors answer: Thank you for your comments. The level of LFB staining and the MBP expression were lower in diabetic mice with ET-1 injection, than non-diabetic mice with ET-1 injection, but with no significance. Therefore, we have revised this conclusion “diabetic mice suffered more severe white matter damage and neurological dysfunction after ET-1 injection than did non-diabetic mice” (line 228-230; line 305-306). According to our results, it is difficult to judge ET-1 induced more sever demyelination or slower the remyelination process. Ischemic stroke leads to white matter injury, inhibiting demyelination or promoting remyelination after white matter injury may have therapeutic value in clinical practice of ischemic stroke. Remyelination is an important repair mechanism triggered after a ischemic stroke-induced white matter injury, but it often fails due to the lack of recruitment of the OPCs to the demyelinated area and the inadequate differentiation of OPCs [1]. In our study, we found that sigma-1 receptor activation could promote OLGs differentiation and OPCs proliferation in mice with diabetes and stroke. We estimate that demyelination and remyelination may be both involved in the toxicity of ET-1 or the neuroprotective of PRE084. The details have been added in Discussion section (line 295-304).

[1]Cheng X, Wang H, Liu C, Zhong S, Niu X, Zhang X, Qi R, Zhao S, Zhang X, Qu H, Zhao C. Dl-3-n-butylphthalide promotes remyelination process in cerebral white matter in rats subjected to ischemic stroke. Brain Res. 2019;1717:167-175. doi: 10.1016/j.brainres.2019.03.017.

2.“Third, long-term PRE084 treatment (for at least 21 d) could prevent diabetes aggravated white matter damage by increasing myelin regeneration and promoting neuroprotection. “  Different oligodendrocyte linage cells number or protein level change are good sign of myelin recovery. Electron microscopy of myelin should be added to give a clear conclusion about myelin recovery. The last thing is that the authors gave PRE084 before ET-1 injection, so the drug may also play a role in protecting OL from apoptosis not only by increasing myelin regeneration. EM of myelin sheath at different conditions and time point will help the authors answer these questions.

The authors answer: Thank you for your comments. With your valuable suggestions, we realized the role of electron microscopy in the evaluation of myelin recovery. In our previous study, PRE084 shows anti-apoptosis effect in mice with cerebral ischemia/reperfusion injury [1]. Therefore, as you suggest, Sigma-1 receptor activation may play protect OLGs from apoptosis not only by increasing myelin regeneration. But, unfortunately, we did not prepare brain tissues for the electron microscope test at the end of the experiment. We will do relevant research in future research to better explain our results. The details have been added in Discussion section (line 232-236).

[1]Zhao X, Zhu L, Liu D, Chi T, Ji X, Liu P, Yang X, Tian X, Zou L. Sigma-1 receptor protects against endoplasmic reticulum stress-mediated apoptosis in mice with cerebral ischemia/reperfusion injury. Apoptosis. 2019 Feb;24(1-2):157-167.

All the lines indicated above are in the revised manuscript.

Thank you for the kind advice.

Sincerely yours,

Dr. Peng Liu

Department of Pharmacology, Shenyang Pharmaceutical University

103 Wenhua Road, Shenyang Liaoning 110016, P.R. China

Tel./Fax: +86-24-23986260

Thank you very much for consideration!

Reviewer 2 Report

The manuscript entitled ‘Sigma-1 Receptor Activation Improves Oligodendrogenesis and Promotes White Matter Integrity after Stroke in Mice with Diabetic Mellitus’ by Song et al. tends to convey the information that activation of Sigma1 receptor (S1R) ameliorated cognitive deficit induced by stroke and diabetes. Authors induced the disease model by combining  vasoconstrictor peptide endothelin-1 (ET-1) induced stroke and high fat diet induced diabetes in C57 mice. A series of studies, including behavioural tests, immunostainings of neurons, OPCs and OLs were performed to confirm that activation of S1R receptor with PRE is protective for the mice in stroke and diabetes. The aim of the study is clear, however, the results are not supporting author’s conclusion completely. The concerns are as following:

Major points:

1. Authors should show the images of brain sections with infarct lesion, at least to show where the lesion is and whether the stroke model is successfully established.

   2.In figure 2, authors showed the immunostaining of Nissil and concluded that S1R activation protects neuronal loss. However, quantification of the neurons are necessary to conclude such statement.

   3. In figure 6, authors used Olig2 to detect oligodendrocytes. However, Olig2 is a lineage marker meaning this antibody can detect OPCs and mature OLs. Since authors claim NG2 cell number is also increased, then the increase of Olig2 can not support author’s conclusion. In addition, Olig2 can be expressed by astrocytes under pathological conditions. Therefore, one should use OL specific marker, Myrf or CC1 or GSTpi, etc. to detect OL density.

  4. In figure 7, authors used NG2 as a marker for OPCs and quantified the area of occupancy to indicate the density of OPCs. However, in the brain, pericytes also express NG2 protein, as well as microglia under pathological conditions. Authors could try PDGFR alpha antibody, which selectively detects OPCs and quantify the PDGFRa+ cell density, instead of fluorescence intensity.

  Minor point:

1.       In general, authors mainly tested hippocampus, partially cortex. Corpus callosum is only tested once for myelin amount. Why is so?

2.       Figure 3 only shows more myelin in the PRE treated mice. But this does not indicate whether this is attribute to the myelin renewal or reduced loss of myelin.

3. The caption for Scheme 1 is wrong. the results show do effect of PRE at 1d time point.

4. In many figures, the legends are missing. For example, in figure 6, authors should label what blue/magenta channel stands for?

5. In figure 7, OPC differentiation is not correct. When they are differentiated, they become oligodendrocytes and do not express NG2.

7. In line 51-52, authors probably meant ‘demyeliantion‘ instead of ‘remyelination‘

8.       Line 53, ‘endogenous’ is not correct. Probably authors meant ‘pre-existing’?

9.       Line 367, why authors used CNPase staining for lesion test?

10.   Line 373-374, what does ‘occurrence’ mean here?

11.   Data are shown in mean +- SD, not SEM, at least for Scheme 1. Please change the figure or the sentence in line 394.

Author Response

Paper: molecules-2039931

Title: Sigma-1 receptor activation improves oligodendrogenesis and promotes white matter integrity after stroke in mice with diabetic mellitus

Thank you very much for your comments on our paper. We have revised our paper according to your comments. All the revisions were emphasized in red.

Reviewer 2:

Major points:

1.Authors should show the images of brain sections with infarct lesion, at least to show where the lesion is and whether the stroke model is successfully established.

The authors answer: Thank you for your comments. The result of the whole brain nissil stain was added as a supplementary result to show the infarct lesion. Because our injection site is the hippocampal CA1 region, so the injury site is mainly concentrated here.

2.In figure 2, authors showed the immunostaining of Nissil and concluded that S1R activation protects neuronal loss. However, quantification of the neurons are necessary to conclude such statement.

The authors answer: Thank you for your comments. We added the nissil cell numbers after ET-1 injection in figure 2.

3.In figure 6, authors used Olig2 to detect oligodendrocytes. However, Olig2 is a lineage marker meaning this antibody can detect OPCs and mature OLs. Since authors claim NG2 cell number is also increased, then the increase of Olig2 can not support author’s conclusion. In addition, Olig2 can be expressed by astrocytes under pathological conditions. Therefore, one should use OL specific marker, Myrf or CC1 or GSTpi, etc. to detect OL density.

The authors answer: Thank you for your comments. The result of Myrf was added in figure 7.

4.In figure 7, authors used NG2 as a marker for OPCs and quantified the area of occupancy to indicate the density of OPCs. However, in the brain, pericytes also express NG2 protein, as well as microglia under pathological conditions. Authors could try PDGFR alpha antibody, which selectively detects OPCs and quantify the PDGFRa+ cell density, instead of fluorescence intensity.

The authors answer: Thank you for your comments. The result of PDGFR alpha was added in figure 9.

Minor point:

1.In general, authors mainly tested hippocampus, partially cortex. Corpus callosum is only tested once for myelin amount. Why is so?

The authors answer: Thank you for your comments. We tested the effect of sigma-1 receptor on cognitive function in mice with diabetes and stroke. Hippocampus and cerebral cortex are closely related to learning and memory. Therefore, we mainly tested these two brain regions.

2.Figure 3 only shows more myelin in the PRE treated mice. But this does not indicate whether this is attribute to the myelin renewal or reduced loss of myelin.

The authors answer: Thank you for your comments. The description has been changed to “Sigma-1 receptor activation prevents the levels of brain demyelination and white matter damage in mice with diabetes and stroke” (line 124-125).

3.The caption for Scheme 1 is wrong. the results show do effect of PRE at 1d time point.

The authors answer: Thank you for your comments. The caption of supplementary fig. 2 has been revised. But there were no significant difference in protein expression among all groups (line 163-166).

4.In many figures, the legends are missing. For example, in figure 6, authors should label what blue/magenta channel stands for?

The authors answer: Thank you for your comments. The caption of fig. 6 and 8 has been revised. In fig. 6, blue channel stands for DAPI, magenta channel stands for Olig2 positive stain. In fig. 8, blue channel stands for DAPI, magenta channel stands for NG2 positive stain (line 184, 211-212)

5.In figure 7, OPC differentiation is not correct. When they are differentiated, they become oligodendrocytes and do not express NG2.

The authors answer: Thank you for your comments. NG2 is expressed by OPCs, not oligodendrocytes. In figure 7, PRE084 significantly decreased the amount of NG2 at 21 d after ET-1 injection. We speculate that OPCs may be differentiated into oligodendrocytes, which leads to the decrease of NG2 positive expression. But, with your reminder, we realize that there are some problems in this conclusion. Therefore, we revised to “PRE084 regulates OPCs proliferation in mice with diabetes and stroke” (line 195, 207-209, 211)

7.In line 51-52, authors probably meant ‘demyeliantion‘ instead of ‘remyelination‘

The authors answer: Thank you for your comments. The “remyelination” has been changed with “demyeliantion” (line 54).

8.Line 53, ‘endogenous’ is not correct. Probably authors meant ‘pre-existing’?

The authors answer: Thank you for your comments. The “endogenous” has been changed with “pre-existing” (line 55).

9.Line 367, why authors used CNPase staining for lesion test?

The authors answer: Thank you for your comments. Our description is wrong. CNpase is a mature OLG marker, not suitable as a marker of brain lesion. The description has been revised (line 404).

10.Line 373-374, what does ‘occurrence’ mean here?

The authors answer: Thank you for your comments. The “occurrence” has been changed with “levels” (line 411).

11.Data are shown in mean +- SD, not SEM, at least for Scheme 1. Please change the figure or the sentence in line 394.

The authors answer: Thank you for your comments. All parameters in fig. 1 has been revised as mean ± SD. And the description of “4.12. Statistical analysis” has also been revised (line 433-434).

All the lines indicated above are in the revised manuscript.

Thank you for the kind advice.

Sincerely yours,

Dr. Peng Liu

Department of Pharmacology, Shenyang Pharmaceutical University

103 Wenhua Road, Shenyang Liaoning 110016, P.R. China

Tel./Fax: +86-24-23986260

Thank you very much for consideration!